# The Effect of Heat Treatment on Damping Capacity and Mechanical Properties of CuAlNi Shape Memory Alloy

**DOI:** 10.3390/ma15051825

**Published:** 2022-02-28

**Authors:** Ivana Ivanić, Stjepan Kožuh, Tamara Holjevac Grgurić, Ladislav Vrsalović, Mirko Gojić

**Affiliations:** 1Department of Physical Metallurgy, Faculty of Metallurgy, University of Zagreb, Aleja Narodnih Heroja 3, 44000 Sisak, Croatia; kozuh@simet.unizg.hr (S.K.); gojic@simet.unizg.hr (M.G.); 2Faculty of Medicine, Catholic University of Croatia, Ilica 242, 10000 Zagreb, Croatia; tamara.grguric@unicath.hr; 3Department of Electrochemistry and Materials Protection, Faculty of Chemistry and Technology, University of Split, Ruđera Boškovića 35, 21000 Split, Croatia; ladislav@ktf-split.hr

**Keywords:** shape memory alloys, heat treatment, microstructure, damping capacity, tensile strength, hardness

## Abstract

This paper discusses the effect of different heat treatment procedures on the microstructural characteristics, damping capacities, and mechanical properties of CuAlNi shape memory alloys (SMA). The investigation was performed on samples in the as-cast state and heat treated states (solution annealing at 885 °C/60′/H_2_O and after tempering at 300 °C/60′/H_2_O). The microstructure of the samples was examined by light microscopy (LM) and scanning electron microscopy (SEM) equipped with a device for energy dispersive spectrometry (EDS) analysis. Light and scanning electron microscopy showed martensitic microstructure in all investigated samples. However, the changes in microstructure due to heat treatment by the presence of two types of martensite phases (β_1′_ and γ_1′_) influenced alloy damping and mechanical properties by enhancing alloy damping characteristics. Heat treatment procedure reduced the alloys’ mechanical properties and increased hardness of the alloy. Fractographic analysis of the alloy showed a transgranular type of fracture in samples after casting. After solution annealing, two types of fracture mechanisms can be noticed, transgranular and intergranular, while in tempered samples, mostly an intergranular type of fracture exists.

## 1. Introduction

The main goal of the application of materials with high damping capacity is a reduction in mechanical vibrations by energy dissipation. For this type of application, materials must also have good mechanical strength, and good electrical and thermal conductivities [1]. However, the damping capacity of shape memory alloys (SMAs) is far greater than that of standard materials [2].

The functional properties of shape memory alloys are influenced by thermoelastic martensitic transformation, which operates in a certain temperature range, depending on the alloy’s chemical composition [3]. These functional properties are affected by the mobile nature of the interfaces (twin boundaries, austenite/martensite (A/M) phase boundaries, different martensite variants) as well as the type of material, grain size, and defects in the structure [4,5].

Copper based SMAs are of interest for investigation due to their high thermal stability. Moreover, they exhibit good damping due to their unique martensitic transition characteristics [3,6].

Depending on alloy composition, applied stress, or temperature, the CuAlNi alloy can transform into different types of martensitic phase, γ_1′_ (2H), β_1′_ (18R1), and α_1′_ (6R). There is a change in martensitic phase due to the variation in compositions (Cu-(11-14)Al-(3-4.5)Ni, wt.%) from the β-β_1′_ transformation to the β-γ_1′_ by aging in the austenite phase [2]. Due to susceptibility to microstructural changes by aging, damping properties are also likely to substantially change [3,7].

The damping capacity, which consists of transferring one form of energy into another (the dissipation of mechanical energy into heat) is a valuable property that characterizes almost all materials including shape memory materials. In shape memory alloys, damping capacity is related to changes during martensitic transformation. The interfaces between A/M phases or between the different variants of martensite influence alloy damping capacity as well as twin boundaries inside the martensitic phase. Although the martensitic transformation has the most significant impact on the functional properties of alloys, other microstructural defects such as dislocations, vacancies, etc. can cause significant changes [8,9].

Hysteresis observed in pseudoelasticity is one of the energy dissipation manifestations [8]. According to the results of Wu et al. [10], in Cu-xZn-11Al (x = 7.0, 7.5, 8.0, 8.5, and 9.0 wt.%), a higher amount of martensite is transformed during martensite transformation, so the alloy exhibits a higher amount of energy dissipation due to higher hysteresis.

For shape memory alloys, it also depends on the difference between the operating and transformation temperatures. In general, three damping regimes can be distinguished in SMAs: (a) in the austenitic phase, the damping capacity is small; (b) an increase in damping capacity for operating below M_f_ temperature; and (c) the damping capacity reaches its maximum by stress induced martensite [2].

In recent research [11,12,13], the focus was primarily on phase field modeling and the simulation of martensite transformation and structural defect interaction under the thermal loading or martensite reorientation and de-twinning process. It can be seen that the evolution of microstructural constituents helps to understand microstructural mechanisms and their overall effect on the optimum performances of alloys. Li et al. [12] explained that not only does the twinning affect alloy damping, but also the reorientation in the martensite microstructure itself.

The aim of this study was to investigate how the changes in microstructure affected by heat treatment procedures influence the Cu-12.8 Al-4.1 Ni shape memory alloy damping and mechanical properties.

## 2. Materials and Methods

The sample of Cu-12.8 Al-4.1 Ni (wt.%) shape memory alloy was prepared in a vacuum induction furnace by the vertical continuous casting technique. The obtained 8 mm diameter rods were mechanically prepared for different stages of investigation. Dimensions of the investigated samples were ø 8 mm × 10 mm. One sample was left in the as-cast state and two samples were subjected to the heat treatment procedure. First, solution annealing was performed at 885 °C with a retention time of 60 min following water quenching. After solution annealing, one sample was subjected to tempering at 300 °C for 60 min following water quenching. The samples for microstructural observation were cut off in the shape of a cylinder, placed in conductive mass, ground, polished, and etched by the procedure explained in our previous work [14]. In order to reveal microstructural constituents, microstructural analysis after etching was performed using a light microscope (LM) OLYMPUS GX 51 with a digital camera and scanning electron microscope TESCAN VEGA TS 5136 MM (SEM) equipped with energy dispersive spectrometry (EDS). Fracture surface morphology was investigated using a scanning electron microscope JEOL JSM 5610 at several different magnifications.

For grain size measurements, the linear intercept procedure was used according to ASTM-E112-13 on optical micrographs at a magnification of 100×. The method involves an actual count of the number of grain boundary intersections by a test line, per unit length of test line. The average value of grain size in the as-cast state sample was calculated according to 512 counted intersections.

Phase transformation temperatures and the alloy’s ability for damping capacity were tested using the dynamic mechanical analysis technique on a TA Instruments device, DMA 983, at a constant frequency of 1 Hz with amplitude of 0.5 mm and at a heating rate of 2 °C min^−1^.

Investigations of mechanical properties were carried out on universal tensile testing machine (Zwick/Roell Z050). Samples were mechanically prepared by turning at the dimensions shown in Figure 1. Tensile testing was performed on three samples for each state at room temperature with a tensile testing rate of 0.001 mm s^−1^.

## 3. Results and Discussion

In order to explain the damping capacity and mechanical properties of the investigated alloy, it is extremely important to examine how process parameters (casting, heat treatment) affect the microstructural characteristics of the alloy.

### 3.1. Microstructural Analysis of CuAlNi Shape Memory Alloy

The results of the light and scanning electron microscopy are presented in Figure 2 and Figure 3, respectively. The martensitic microstructure was clearly visible in all of the samples, even in the as-cast state sample (Figure 2a and Figure 3a). Often, during solidification of Cu-based SMAs, a residual austenitic phase or brittle γ_2_ phase (Cu_9_Al_4_ phase) can be found, which strongly influences an alloy’s functional properties in the as-cast state [4]. Hence, the heat treatment procedure cannot be avoided to improve the alloy’s properties. The heat treatment process by solution annealing the CuAlNi SMA was carried out to achieve a fully martensitic microstructure. In addition to the formation of the martensitic phase from the initial austenitic (β) phase, there was a change in grain size that depends on the conditions of the heat treatment process (heat treatment temperature, retention time at certain temperature, and the choice of cooling medium).

Changes in the CuAlNi alloy microstructure vary depending on the heat treatment process [15]. In Figure 2b,c, we see the martensitic microstructure after solution annealing and tempering. In the micrographs obtained by light microscopy, it can be observed that during solution annealing and tempering, the grain size increased within the microstructure compared to the as-cast state sample. Grain size is an extremely important characteristic of the microstructure because other mechanical and functional properties of the alloy depend on it. The disadvantage of shape memory alloys is the coarse-grained microstructure, which negatively affects the behavior of the alloy. Grain size in the as-cast state was 158.76 µm (average value). Average value of the grain size in the as-cast state was obtained by the linear intercept procedure through 512 measurements of line intercepting points. ASTM grain size (G) was 2.02 and the number of grains per unit area (N_A_) was 31.62 grains per square millimeter.

In addition, the grain sizes changed from the edge toward the middle of the rod because of solidification (i.e., directed heat dissipation where the rod is in contact with the crystallizer, small grains appear). After solution annealing and tempering, larger uniform grains in the microstructure could be observed, even in the order of 1 mm (Figure 2b,c). Kök et al. [16] investigated the thermal stability of the quaternary CuAlNiTa alloy and concluded that the heat treatment procedure enhanced grain size. Furthermore, Xi et al. [13] performed a phase field study of the grain size effect on the thermomechanical behavior of a NiTi shape memory alloy thin film and concluded that the grain size had an inhibiting effect on the temperature induced martensitic transformation.

The martensitic microstructure was confirmed on all samples and by scanning electron microscopy (Figure 3). Different orientations of martensite needles within a single grain can be explained by the nucleation of groups of martensitic needles at numerous sites within the grain and the creation of a local stress within the grain that allows for the formation of multiple groups of differently oriented needles. Martensite originated primarily as needle martensite. In some samples, after solution annealing and tempering (Figure 3b,c), the V-shape of martensite could be observed. The morphology of the resulting martensitic microstructure was a typical self-accommodating zig-zag morphology, which is primarily characteristic of β_1′_ martensite in CuAlNi shape memory alloys [17,18].

It has also been confirmed in the literature [18,19] that different variants are characteristic of self-accommodating martensite in CuAlNi alloys and that these are most often the two types of heat-induced martensite (β_1′_ and γ_1′_ (18R and 2H)) that coexist in the microstructure. The β_1′_ martensite appears as lath type martensite with thin plates (needle-like morphology) and γ_1′_ is thicker with plate-like morphology (coarse variants) [5]. It has been reported that the γ_1′_ martensite with a 2H structure in the CuZnAl alloy has abundant movable twins [10].

### 3.2. Results of Dynamic-Mechanical Analysis

Dynamic-mechanical analysis (DMA) is a technique of the thermal analysis of materials by which we can monitor the response of materials to cyclic loading during the controlled heating of materials at different temperature, time, frequency, stress, atmosphere, or a combination of these parameters. The sinusoidal cyclic stress of the material results in deformations that change sinusoidally with time at the same frequency [20]. The storage modulus (E’) is related to the properties of the elastic component and is proportional to the stored energy, which is returned as mechanical energy during periodic deformation. The stress component bound to the viscous component is determined by the size of the loss modulus (E”) proportional to the lost mechanical energy in the form of heat. The phase shift angle is given by the ratio of the loss module (E”) and the storage module (E’) and is a measure of the energy loss in the material due to viscous friction [21].

The results of the dynamic-mechanical analysis measurements are shown in Figure 4, Figure 5 and Figure 6. The storage modulus (E’), loss modulus (E”), and mechanical damping parameter (tan δ) are shown as a function of temperature during heating.

Dynamic-mechanical analysis showed the dependence of the tangent of the phase shift angle, which is a measure of the energy loss in the material (tan δ), the storage modulus (E’), and the loss modulus (E”) with temperature. The DMA spectra of the investigated samples indicated that the highest temperature of austenitic transformation (A_s_ = 220 °C, A_f_ = 250 °C) was in the samples in the as-cast state, which is in accordance with the obtained results of the differential scanning calorimetry reported in our previous work [22]. Temperatures of A_s_ and A_f_ were higher than the DSC results (A_s_ = 179 °C, A_f_ = 212 °C), which is a common measurement difference between the two techniques (DSC and DMA), reported in the research by Graczykowski et al. [23]. The maximum intensity of the loss modulus (E”) and the change in the storage modulus (E’) indicated a slightly lower austenitic transformation temperature in the solution annealed and tempered alloy and the lowest temperature in the solution annealed alloy at 885 °C/60′/H_2_O. The values of the storage modulus were highest in the martensitic structure of the as-cast alloy and decreased with the obtained heat treatment procedure, which correlated with the tests of the mechanical properties and tensile strength (Figure 7, Table 1). The lowest values of tan δ and E” could be observed in the sample in the as-cast state. The phase shift angle tan δ increased for the solution annealed and tempered state, and it can be concluded that the energy loss ability in the material was higher for the samples after solution annealing and tempering. In addition, the loss modulus (E”) increased for the solution annealed state sample, and was slightly lower for the tempered state sample. Since heat treatment leads to changes in the microstructure, it can be concluded that the newly formed intermediate boundaries of martensite (β_1′_ and γ_1′_ martensite) favorably affect the ability for damping of the CuAlNi shape memory alloy. Ursanu et al. investigated CuAlMn shape memory alloys by DMA and discovered that after heat treatment process by aging above 300 °C, the alloy’s damping capacity decreased due to formations of brittle γ_2_ precipitates, which restricted the mobility of the martensite interfaces [24].

The difference in tan δ values between the solution annealed and tempered state was also noticed by Suresh and Ramamurty [3] (80% higher tan δ in the solution annealed sample). Damping capacity of the shape memory alloys can be significantly altered by an artificial aging process. Shape memory alloys have a high damping capacity due to the ability to move the austenite/martensite interface during the phase transformation. However, it is difficult to move the interfacial boundary due to the precipitation of the low-temperature γ_2_ phase and unfavorable influence on alloy damping capacity [3].

Chang [25] investigated the damping capacity by dynamic mechanical analysis on a Cu-X% Al-4% Ni (X = 13.0–14.1 wt.%) SMA. He found that an alloy with 14% Al, thanks to the high concentration of moving twins interfaces in γ_1′_ (2H) martensite, satisfied the application in which the vibration damping property is required under isothermal conditions.

### 3.3. Results of Mechanical Properties of the CuAlNi Shape Memory Alloy

The investigated CuAlNi shape memory alloy showed satisfactory results of stress, strain, and hardness in the as-cast state (Table 1, Figure 7). With heat treatment procedure, the values of tensile strength and strain decreased, while the hardness increased significantly. The reason for such behavior can be explained by changes in the obtained microstructure and grain size. Although the mechanical properties notably decreased after heat treatment, the development of favorable martensite phases produced better damping properties. Two types of martensite can be found in this alloy, β_1′_ and γ_1′_, respectively [26]. Appearance of γ_1′_ martensite improves the damping capacity of the alloy due to movement of twin boundaries in the martensite phase and higher hysteresis in comparison to β_1′_ martensite [3]. Improvement in the mechanical properties can also be achieved by grain refinement procedures (microalloying or rapid solidification techniques) in order to affect or limit the movement of internal dislocations in the alloy [27].

Significant changes in fracture mechanism could be noted by SEM fractography analysis, Figure 8. In the as-cast state, a mainly transgranular type of fracture could be seen including areas with small and shallow dimples, assuming that certain plasticity in the sample had appeared. After solution annealing, the appearance of the intergranular type of fracture was visible, and after tempering at 300 °C/60′/H_2_O, an almost completely intergranular type of fracture appeared. The degradation of fracture mechanisms, or shift from a transgranular type of fracture to intergranular type after heat treatment, can be explained by the alloy’s high elastic anisotropy [28,29] and large grain size [30]. Stress level concentration is high at large grain boundaries and nucleation of the cracks appears specifically in these places due to low cohesion. It was shown that because of the low ductility of the CuAlNi SMA, cracking of the samples occurred even during elastic deformation. This investigation is in accordance with the results that Pushin et al. [31] obtained in the studies of different NiTi-based and Cu-based shape memory alloys by confirming that large grains in the NiTi alloy revealed brittle fracture as well as classic brittle fracture along the grain boundaries in the investigated CuAlNi SMA.

## 4. Conclusions

An investigation of the microstructural and dynamic-mechanical properties of the CuAlNi shape memory alloy was carried out. From the detailed analysis, the following conclusions can be drawn:Microstructural analysis by light microscopy and scanning electron microscopy revealed the martensitic phase in the as-cast, solution annealed, and tempered state samples. The appearance of two types of martensite phases (β_1′_ and γ_1′_) was noted.Dynamic-mechanical analysis showed a higher damping capacity for samples after solution annealing and tempering. The newly formed interfacial boundaries of martensite (β_1′_ and γ_1′_ martensite) favorably affected the damping properties of the CuAlNi shape memory alloy.Results of the mechanical properties showed degradation of stress, strain, and hardness in the heat treated states due to large grain size, which appeared after the heat treatment process.Fractographic analysis showed a transgranular type of fracture in the as-cast state sample, transgranular and intergranular type of fracture in the solution annealing state, and mostly intergranular type of fracture in the tempered state sample. It can be concluded that the brittleness of the alloy arose from this heat treatment process.

## Figures and Tables

**Figure 1 materials-15-01825-f001:**
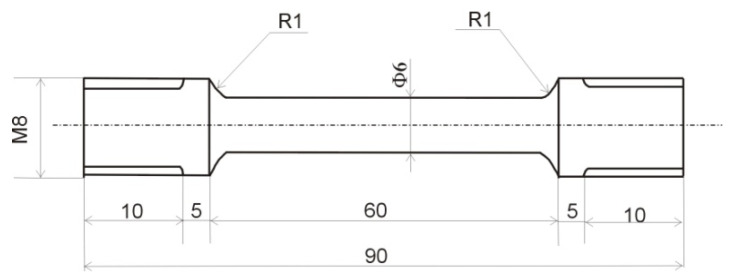
Schematic illustration of the tensile test sample.

**Figure 2 materials-15-01825-f002:**
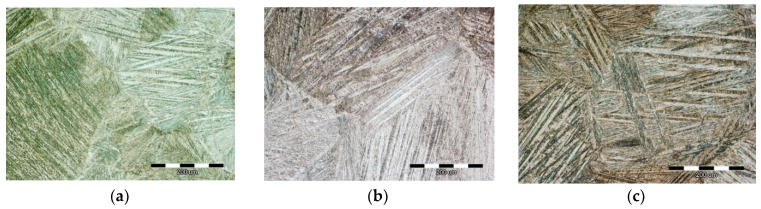
Light micrographs of the CuAlNi shape memory alloys in the as-cast state (**a**), after solution annealing at 885 °C/60′/H_2_O (**b**), and after solution annealing at 885 °C/60′/H_2_O and tempering at 300 °C/60′/H_2_O (**c**), magnification 200×.

**Figure 3 materials-15-01825-f003:**
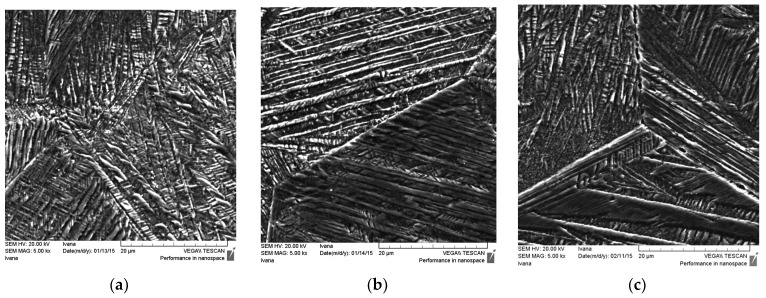
SEM micrographs of the CuAlNi shape memory alloys in the as-cast state (**a**), after solution annealing at 885 °C/60′/H_2_O (**b**), and after solution annealing at 885 °C/60′/H_2_O and tempering at 300 °C/60′/H_2_O (**c**).

**Figure 4 materials-15-01825-f004:**
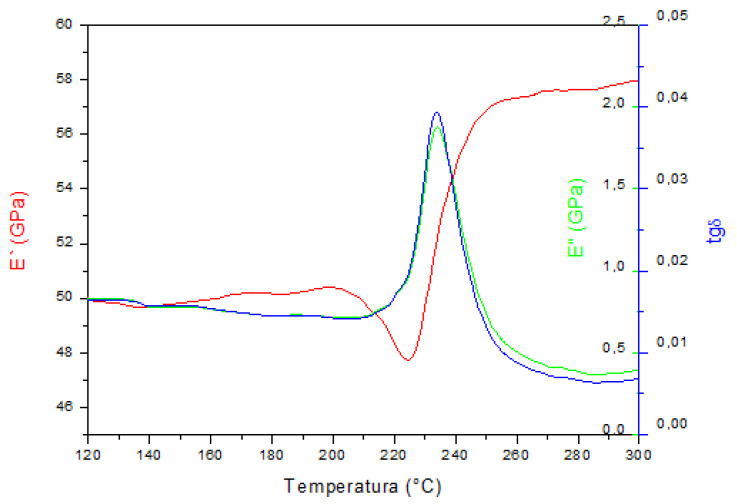
DMA spectrum of the CuAlNi shape memory alloy in the as-cast state.

**Figure 5 materials-15-01825-f005:**
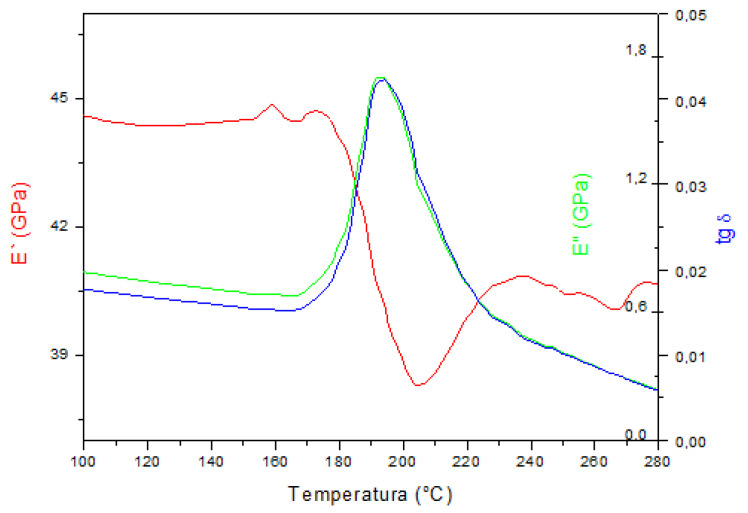
DMA spectrum of the CuAlNi shape memory alloy after solution annealing at 885 °C/60′/H_2_O.

**Figure 6 materials-15-01825-f006:**
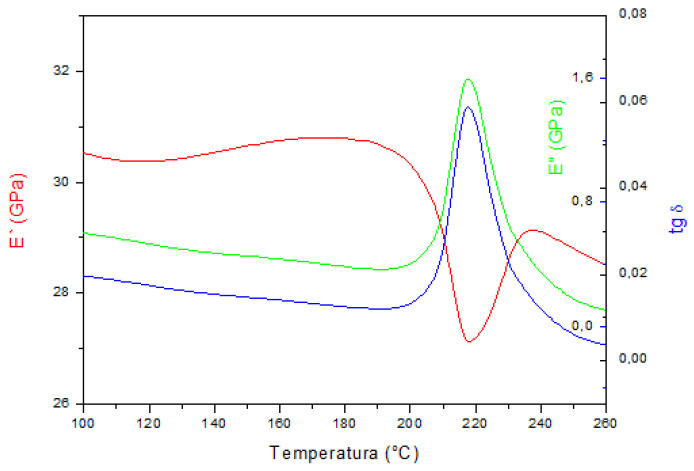
DMA spectrum of the CuAlNi shape memory alloy after solution annealing at 885 °C/60′/H_2_O and tempering at 300 °C/60′/H_2_O.

**Figure 7 materials-15-01825-f007:**
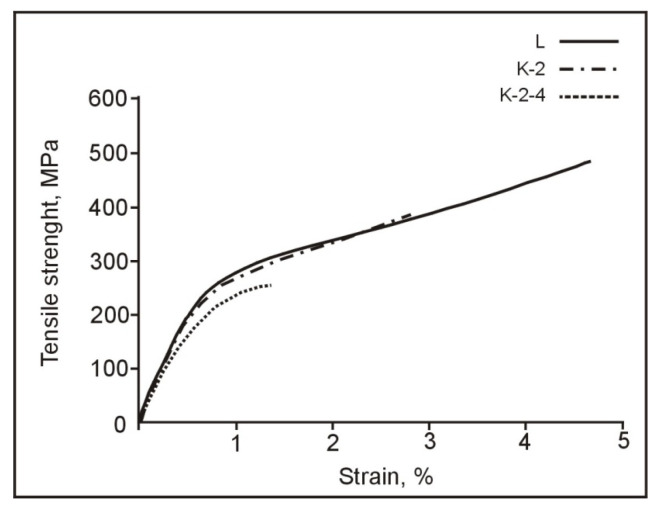
Stress vs. strain in the CuAlNi SMA.

**Figure 8 materials-15-01825-f008:**
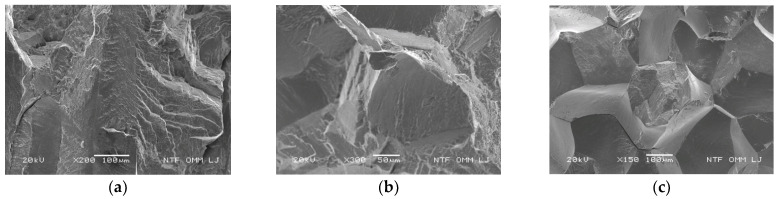
Fracture surface morphology after tensile testing in the as-cast state (**a**), after solution annealing at 885 °C/60′/H_2_O (**b**), and after solution annealing at 885 °C/60′/H_2_O and tempering at 300 °C/60′/H_2_O (**c**).

**Table 1 materials-15-01825-t001:** Mechanical properties of the CuAlNi alloy after casting and heat treatment.

Sample	Tensile Strength, MPa	Elongation, %	Hardness, HV1
L (as-cast state)	475.5 ± 9.3	4.78 ± 0.28	344.0 ± 18.4
K-2 (885 °C/60′/H_2_O)	367.5 ± 48.8	2.72 ± 0.46	480.0 ± 25.0
K-2-4 (885 °C/60′/H_2_O + 300 °C/60′/H_2_O)	241.7 ± 11.9	1.36 ± 0.18	483.0 ± 14.4

## Data Availability

The data presented in this study are available on request from the corresponding author.

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
