# Peer review of "The Effect of Heat Treatment on Damping Capacity and Mechanical Properties of CuAlNi Shape Memory Alloy"

_materials, 2022, doi:10.3390/ma15051825_

Round 1
Reviewer 1 Report
The authors investigated the damping capacity and mechanical property of CuAlNi SMA by studying the martensitic microstructure as a result of heat treatment. The topic of this work fits the scope of the journal very well. The manuscript contains quite interesting contents related to thermodynamic behavior of the material and the microstructures. Some issues are suggested to be addressed:
1. The writing and English need to be polished. Grammatical problems are occasionally seen in the texts, e.g., “Often, during solidification of Cu-based SMAs can be find the residual austenitic phase ...” and others.
2. For the stress-strain behavior presented in Fig. 6. Was true strain calculated and used or just the engineering strain presented in the figure?
3. This is an experimental work which characterized the macroscopic property of CuAlNi SMA by examining the martensitic microstructure of the materials. For instance, the authors stated that martensites are able to improve damping capacity of the ally through the movement of the twin boundaries. Therefore, it is recommended that the authors do a better literature review in the introduction and add relevant and updated references regarding the advanced studies of twin boundary movement and phase transformation in SMAs, e.g., https://doi.org/10.1007/s00707-020-02613
https://doi.org/10.1007/s00707-021-03067-5
https://doi.org/10.1007/s00707-021-03074-6
This work may be re-considered for publication after the above issues are properly addressed.
Author Response
Please, see the attached response.

Reviewer 2 Report
The microstructural characteristics, damping capacity and mechanical properties of the CuAlNi shape memory alloy (SMA) had been studied under the condition of different heat treatment. The research could help to optimize the heat treatment process of this alloy in factory. From my point of view, the work is well-done and provides interesting results to the production. But the content of this manuscript isn’t much enough, so it could be published after the content had been added. And there is also a mistakes in section “1 Introduction” line 9, the word “autenite” should be “austenite”.
Reviewer 3 Report
1. In [2. Materials and Methods], the details of the tensile experiment should be introduced, such as sample shape, size, surface polishing state, and tensile rate.
2. In [3.1. Microstructural analysis of CuAlNi shape memory alloy], the author introduced that [Grain size in the as cast state was 158.76 µm (average value).], the data here is not rigorous, the author should introduce about crystal Grain statistics is to count the number of optical microscope photos, whether the statistics are all complete grains, at the same time, error bars should also be added.
3. In [Table 1. Mechanical properties of the CuAlNi alloy after casting and heat treatment], the test of mechanical properties is not rigorous enough. Each set of samples should be at least 4 samples, and error bars should be added. In addition, the topography of the tensile fracture should also be provided
Reviewer 4 Report
The current manuscript discusses the effect of heat treatment on the damping capacity and mechanical behavior of CuAlNi shape memory alloy. The topic is widely discussed in the literature and the authors did not present the results propertly. The reviewer detected a high plagiarism percentage in the introduction as shown in the attached file. General comments include the following:
- The first affiliation in the address section is incomplete.
- High plagiarism percentage was detected in the introduction section.
- The objectives of the work are not clearly mentioned.
- The materials and methods section must be improved.
- The results are not well presented and the discussion is insufficient in many locations.
- The conclusions kust be improved
Specific comments can be found in the attached file. Hoping these comments to improve the quality of the submitted manuscript.

Reviewer 5 Report
The article does not present a systematic scientific study, but in its current form presents fragmented results from several experiments and is not very useful for the readers. The length of the paper is not enough, also the list of references is not adequate.
I recommend rejecting the paper.
Round 2
Reviewer 1 Report
The authors have properly addressed my comments. I recommend it for publication.
Reviewer 4 Report
The authors did not highlight the changes in the revised manuscript and the coverletter does not cover the inquiries properly. Furthermore, the contents are still fragmented. It is highly recommended to resonstruct the manuscript with more scientific soumd results.
Reviewer 5 Report
Remarks to the authors
1. It is unclear what is changed in the paper. The authors didn’t mark or described the changes.
2. Figure 1 is unnecessary.
3. The obtained results are represented graphically, but are not adequately described and studied.
4. Some important papers are not reviewed in the Introduction section.
5. There are many self-citations;
6. It is unclear whether this alloy exhibits hysteresis behavior;
7. The length of the paper is not enough;
8. Although the issue of mechanical and damping behaviour is fundamental to the article, it has not been sufficiently studied. I believe that there is nothing new in the article.
I confirm my initial decision – reject the paper.